# Pro-Inflammatory Cytokines and Interferon-Stimulated Gene Responses Induced by Seasonal Influenza A Virus with Varying Growth Capabilities in Human Lung Epithelial Cell Lines

**DOI:** 10.3390/vaccines10091507

**Published:** 2022-09-09

**Authors:** Alfredo A. Hinay, Sosuke Kakee, Seiji Kageyama, Akeno Tsuneki-Tokunaga, Waldy Y. Perdana, Yui Akena, Shota Nishiyama, Kyosuke Kanai

**Affiliations:** Division of Virology, Department of Microbiology and Immunology, Faculty of Medicine, Tottori University, 86 Nishi-cho, Yonago 683-8503, Japan

**Keywords:** influenza A virus, growth capability, interferon-β, interferon-stimulated genes

## Abstract

In a previous study, we described the diverse growth capabilities of circulating seasonal influenza A viruses (IAVs) with low to high viral copy numbers in vitro. In this study, we analyzed the cause of differences in growth capability by evaluating pro-inflammatory cytokines (TNF-α, IL-6, IFN-β) and antiviral interferon-stimulated genes (ISG-15, IFIM1, and TRIM22). A549 cells (3.0 × 10^5^ cells) were inoculated with circulating seasonal IAV strains and incubated for 6 and 24 h. In cells inoculated for 6 h, IAV production was assessed using IAV-RNA copies in the culture supernatant and cell pellets to evaluate gene expression. At 24 h post-infection, cells were collected for IFN-β and ISG-15 protein expression. A549 cells inoculated with seasonal IAV strains with a high growth capability expressed lower levels of IFN-β and ISGs than strains with low growth capabilities. Moreover, suppression of the JAK/STAT pathway enhanced the viral copies of seasonal IAV strains with a low growth capability. Our results suggest that the expression of ISG-15, IFIM1, and TRIM22 in seasonal IAV-inoculated A549 cells could influence the regulation of viral replication, indicating the existence of strains with high and low growth capability. Our results may contribute to the development of new and effective therapeutic strategies to reduce the risk of severe influenza infections.

## 1. Introduction

The current circulating seasonal influenza A virus (IAV) subtypes H1N1 and H3N2 are the most common causes of pneumonia-related morbidity in developed countries, and mortality can be much higher during pandemics [1,2]. The clinical severity of an influenza virus infection can range from asymptomatic to symptomatic, and it has been shown that a higher viral load is associated with more severe cases of the disease and a high probability of viral transmission and spread [3,4]. Moreover, one of the probable factors affecting viral load is the intrinsic growth capabilities of the virus [5].

Our previous studies demonstrated that emerging pandemic viruses consist of only highly replicative strains, but more slowly replicating strains were observed in the following seasons [5,6]. We hypothesized that the difference in growth capability could be due to the immune response of the human host, and that understanding how viral load and host immune responses are related is critical for identifying disease-specific markers that may help predict disease prognosis.

In response to viral infection, the production of pro-inflammatory cytokines induces the binding of TNF-α, IL-6, and IFN-β to their receptors, followed by activation of the JAK/STAT pathway to initiate the transcription of hundreds of IFN-stimulated genes (ISGs) [7,8,9]. ISGs eliminate the virus and prevent its spread by promoting an antiviral state in neighboring cells, thereby limiting disease severity [10]. In the context of influenza virus infection, ISGs, specifically interferon-stimulated gene 15 (ISG-15), interferon-induced transmembrane protein 1 (IFITM1), and tripartite motif-containing 22 (TRIM22), target genomic replication, viral entry, and inhibition of proteasomal degradation, respectively [11,12,13,14,15,16,17]. However, given the critical role of pro-inflammatory cytokines, specifically IFN-β and ISGs as the first line of defense against IAV infection, the virus has developed strategies to block the IFN-induced JAK/STAT pathway at different levels by using several viral factors through various mechanisms to increase its replication efficiency [7,18]. Among IAV viral proteins, non-structural protein 1 (NS1) has been identified as the main type I IFN antagonistic factor [19].

Several groups have investigated the contribution of NS1 protein mutations to viral replication in IAVs [20,21,22]. Manipulation of a single amino acid substitution at position 45 (G to R; G45R) in a mouse-adapted laboratory strain of influenza (PR8) can inhibit IFN production, leading to enhanced viral replication and an increased viral load [22]. In contrast, manipulation of a single amino acid substitution at position 64 (I to T; I64T) decreases its ability to inhibit interferons, leading to decreased viral replication [23,24,25]. While the IAV NS1 protein is well-studied for its anti-type I IFN function, little is known about the antiviral ISGs response in circulating seasonal IAV strains and its possible contribution to growth capability. In this study, we highlighted that inhibition of the JAK/STAT pathway permits seasonal IAV strains with a low growth capability to enhance their viral copies by suppressing ISG-15, IFITM1, and TRIM22. This finding suggests that the differences in the growth capability of seasonal IAV strains may be due to the regulation of antiviral ISGs.

## 2. Materials and Methods

### 2.1. Cells and Viruses

A549 cells (human lung epithelial cells) were cultured in complete Dulbecco’s modified Eagle’s medium (DMEM; FUJIFILM Wako, Osaka, Japan) supplemented with 10% fetal bovine serum (FBS; Gibco, Waltham, MA, USA) and penicillin–streptomycin at 37 °C in an atmosphere of 5% CO_2_.

From 21 seasonal influenza A virus culture-positive nasal swabs and nasopharyngeal aspirate samples collected from hospitals in Tottori Prefecture (Tsuchie Internal Medicine and Pediatric Clinic, Sakaiminato; Department of Pediatrics, Tottori Prefectural Kousei Hospital, Kurayoshi; Kasagi Children’s Clinic for Health Service, Yonago; and Tanaka Pediatric Clinic, Tottori, Japan), 4 and 3 seasonal IAV-representative strains with high and low growth capabilities with a detectable number of copies in RT-PCR after 6 h post-infections were evaluated for anti-inflammatory cytokines and ISG expression. A reference strain (A/Puerto Rico/8/1934(H1N1), PR8) and four epidemic strains (A/Tottori/ST215/2009(H1N1), A/Tottori/ST1349/2014(H3N2), A/Tottori/ST488/2014(H3N2), and A/Tottori/ST1350/2013(H1N1)) with a high growth capability and three epidemic strains (A/Tottori/TT039/2011(H1N1), A/Tottori/ST1705/2014(H3N2), and A/Tottori/ST777/2011(H3N2)) with a low growth capability were prepared and stored in a deep freezer (−80 °C) until ready for use.

### 2.2. Virus Infection of Cells

A549 cells (3.0 × 10^5^) were washed twice with phosphate-buffered saline (PBS) and inoculated with the seasonal IAV strains. Briefly, an aliquot (500 µL) of 10^7^ viral copies was inoculated into the cells. After 1 h of inoculation, the cells were washed twice with PBS, and maintained in Dulbecco’s modified Eagle’s complete medium supplemented with 5 µg of trypsin (Difco™ Trypsin 250, Becton Dickinson, Tokyo, Japan) per ml, 0.2% heat-inactivated bovine serum albumin, 4 mM L-glutamine, 200 units of penicillin G per ml, and 100 μg of streptomycin per ml, at 34 °C in a 5% CO_2_/95% air atmosphere. The cells were further incubated for 6 and 24 h. Cells were inoculated for 6 h, and the culture supernatants were subjected to viral load and cell pellets for gene expression evaluation. At 24 h post-infection, cells were collected for IFN-β and ISG-15 protein expression.

### 2.3. RNA Extraction and Polymerase Chain Reaction

Influenza RNA in the culture supernatant (140 µL) was extracted using a QIAamp Viral RNA Mini Kit (QIAGEN, Tokyo, Japan) and subjected to quantitative reverse transcription PCR. The amount of influenza RNA was assayed using a Luna Universal One-Step qRT–PCR kit (New England Biolabs, Tokyo, Japan) and specific primers (FLUA-MAT-F/R) of the matrix gene of influenza A (Table 1). The PCR signal was assessed by relating it to a standard curve (encompassing 10^2^ to 10^9^ copies/reaction of in vitro transcribed RNA transcript). RNA transcripts were prepared in-house using cloning constructs of FLUA-MAT-F and FLUA-MAT-R PCR products, and in vitro transcribing using Riboprobe^®^ Combination Systems (Promega, Madison, WI, USA). All analytical runs were performed in triplicate with a no-template control (NCT).

### 2.4. Expression of Pro-Inflammatory Cytokines (TNF-α, IL-6, and IFN-β) and ISGs (ISG-15, IFIM1, and TRIM22)

Total RNA was isolated using a RNeasy Mini Kit (Qiagen, Hilden, Germany), according to the manufacturer’s protocol. RNA quality was evaluated using a NanoDrop ND-1000 spectrophotometer (Thermo Fisher Scientific, Waltham, MA, USA). cDNA was synthesized using a Superscript III cDNA synthesis kit (Invitrogen, Tokyo, Japan), and gene expression was quantified on the StepOne Real-Time PCR System (Applied Biosystems, Waltham, MA, USA) using the SYBR green method. Predesigned primers for the expression analysis are listed in Table 1. The reference gene 18S rRNA was experimentally validated in A549 cells prior to gene expression evaluation. Gene expression was normalized to the reference gene 18S rRNA and graphed as fold change relative to the untreated controls using the 2^−∆∆CT^ method [26,27]. All analytical runs were performed in triplicate with a no-template control (NCT).

### 2.5. IFN-β and ISG-15 Protein Production

Cell lysis buffer (Wako, Fujifilm, Tokyo, Japan) supplemented with protease inhibitor cocktail set I (Wako, Fujifilm, Tokyo, Japan) was used to prepare the cell lysate from 24 h-inoculated low-growth-capability strains for A549 cells. The concentrations of human IFN-β (DuoSet, R&D Systems, Inc. Minneapolis, MN, USA) and ISG-15 (Circulex, MBL Life Sciences, Tokyo, Japan) were measured using an ELISA kit according to the manufacturer’s instructions.

### 2.6. JAK Inhibition

A549 cells were treated with 5 μM pyridone 6 (2-(1,1-dimethylethyl)-9-fluoro-1,6-dihydro7H-benz[h]imidazo[4,5-f] isoquinolin-7-one (CMP6) (Funakoshi Frontiers in Life Science, Tokyo, Japan) and Janus-associated kinase inhibitor 1 (JAK-1) [28]. The JAK inhibitor was prepared as a 10 mM stock concentration in DMSO. Following pretreatment of cells with 5 μM JAK inhibitor and DMSO as a control for 24 h, the medium was removed, and the cells were rinsed with PBS before infection with low-growth -capability IAV strains. At 6 h post-infection, ISG-15, IFITM1, and TRIM22 expression levels were quantified from total RNA.

### 2.7. NS1 Sequencing

Total RNA was extracted using a QIAamp Viral RNA Mini Kit (QIAGEN, Tokyo, Japan). Eight RNA segments of the influenza A virus were simultaneously reverse-transcribed and amplified using the SuperScript™III Step RT-Polymerase Chain Reaction (PCR) System (Life Technologies, Tokyo, Japan) with the MBTuni-12/MBTuni-13 primer pair [5,29]. The first-round PCR products were subjected to a second PCR to amplify NS1 using A-NS-M13 primers (Table 1). The amplified NS1 complete gene sequences were determined using a BigDye^®^ Terminator v3.1 Cycle Sequencing Kit((Thermo Fisher scientific, Waltham, MN) in accordance with the manufacturer’s instructions (Life Technologies) and FASMAC (Atsugi City, Japan) to analyze the data.

### 2.8. Statistical Analysis

Quantitative reverse transcription PCR (qRT-PCR) data were statistically analyzed using the 2^−∆∆CT^ method [26,27]. The Shapiro–Wilk test for normality was performed to ensure a normal distribution prior to parametric analysis using a two-tailed *t*-test. Statistical analysis of viral copies was performed using a two-tailed *t*-test in GraphPad Prism software (version 8.0). *p* values of ≤ 0.05 are given one asterisk (*).

## 3. Results

### 3.1. IAV Showed Diverse Growth Capabilities In Vitro

The 21 seasonal influenza A virus culture-positive nasal swab and nasopharyngeal aspirate samples collected from hospitals in Tottori Prefecture showed diverse growth capabilities in vitro using A549 human lung epithelial cells (Figure 1A). Four strains with the highest viral copy numbers (*n* = 4) and three strains with the lowest viral copy numbers (*n* = 3) were selected to evaluate TNF-α, IL-6, IFN-β, ISG-15, IFITM1, and TRIM22 expression levels at 6 h post-infection.

### 3.2. IFN-β and Antiviral ISG Expression Levels Increased in A549 Cells Inoculated with Low-Growth-Capability IAV Strains

To gain insight into the differences in the growth capabilities of seasonal IAV strains, the gene expression of pro-inflammatory cytokines (TNF-α, IL-6, and IFN-β) and antiviral ISGs (ISG-15, IFITM1, and TRIM22) was determined using A549 cells. We obtained four and three seasonal IAV subtypes, A(H1N1) and A(H3N2), with high and low growth capabilities, respectively. The seasonal IAV strains induced gene expression of TNF-α and IL-6 but did not show significant differences (Figure 1B,C) when compared between high- and low-growth-capability strains. However, IFN-β (*p* = 0.0170), ISG-15 (*p* = 0.0378), IFITM1 (*p* = 0.0004), and TRIM22 (*p* = 0.0020) showed significant differences between the high- and low-growth-capability strains (Figure 1D–G). These results suggest that in strains with a low growth capability, the expression of IFN-β and ISGs were parallel; high expression of IFN-β upregulated the expression of antiviral ISGs. In contrast, the expression of IFN-β and ISGs in IAVs with a high growth capability was downregulated (Figure 1D–G).

### 3.3. IFN-β and ISG-15 Protein Production

Seasonal IAVs with low growth capabilities showed increased IFN-β expression. Among antiviral ISGs, ISG-15 showed the highest expression level. The main limitation of using a cell line in determining relative gene expression is that it only indicates the expression of transcriptionally regulated genes at the mRNA level, and provides no evidence as to whether the specific protein is expressed. To determine whether the difference in growth capability was due to the expression of IFN-β and ISG-15 in A549 cells, protein expression levels were confirmed by ELISA at 24 h post-infection.

IFN-β protein expression at 24 h post-infection was significantly increased (*p* = 0.0067) in IAV strains with a low growth capability compared with those with a high growth capability (Figure 2A). ISG-15 protein expression at 24 h post-infection also showed the same pattern (*p* = 0.0038) (Figure 2B). These results showed a significant difference in IFN-β and ISG-15 production when A549 cells were inoculated with low-growth-capability strains compared with high-growth-capability strains.

### 3.4. JAK/STAT Pathway Inhibition Increased the Susceptibility of A549 Cells to Seasonal IAV Strains with Low Growth Capability

To assess the role of ISGs (ISG-15, IFITM1, and TRIM22) in the growth capabilities of seasonal IAVs, A549 cells were treated with 5 μM pyridine 6 (P6), a pan-JAK inhibitor known to inhibit ISG expression [28,30]. As expected, the expression levels of ISG-15, IFITM1, and TRIM22 were significantly reduced in the P6-treated A549 cells (Figure 3A–C). The suppression of viral ISGs in A549 cells using P6 resulted in a significant increase in viral copies (*p* < 0.05) at 6 h post-infection with seasonal IAV strains with a low growth capability (Figure 3D).

### 3.5. NS1 Mutations Are Present in the Circulating Seasonal IAV Strains

Comparative analyses of various NS1 point mutations in wild-type (WT) and mutant strains have been shown to influence IAV replication, which has been linked to the growth capability of the virus. Here, we checked the NS1 sequences of each seasonal influenza A strain and compared them with existing NS1 mutations that may influence the viral load (Table 2). Notably, all strains with a high growth capability had the G45R and K66E combination mutations. Alternatively, strains with a low growth capability had the I64T and E55X combination mutations.

## 4. Discussion

The growth capability of the influenza A virus is influenced by the interactions between the virus and the host’s regulation factors, with IFN-β and antiviral ISGs being the primary mediators of the host’s innate immune response. These molecules are induced following viral infection and establish an antiviral state in both infected and neighboring cells by stimulating the expression of antiviral genes, known as interferon-stimulated genes (ISGs) [31]. As viral replication is necessary for disease pathogenesis in IAV infections, information on the correlation between viral load and antiviral ISGs can provide answers to the differences in the growth capabilities of seasonal IAV strains. In this study, the differences in the growth capability of seasonal IAVs were dependent on the expression of IFN-β and antiviral ISGs (ISG-15, IFITM1, and TRIM22) in A549 cells. Influenza A virus strains with high growth capabilities are associated with low gene expression of IFN-β and antiviral ISGs. Interestingly, strains with a low growth capability showed an inverse pattern of IFN-β and ISG expression.

In vitro studies have reported that A549 cells inoculated with IAVs express lower levels of IFN-β, demonstrating the capability of viral factors to inhibit the expression of IFN-β [18,32,33,34,35]. However, as described above, we found that circulating IAVs with a low growth capability significantly upregulated IFN-β expression and induced antiviral ISG (ISG-15, IFITM1, and TRIM22) production. Remarkably, among the pro-inflammatory cytokines (TNF-α, IL-6, and IFN-β), only IFN-β showed gene expression differences in seasonal IAV strains with high and low growth capabilities. This suggests that the difference in growth capabilities is due to the strain-dependent regulation of IFN-β and antiviral ISGs. This evidence provides new insights into the mechanism of seasonal IAV replication, as a low growth capability and activation of the IFN-β and JAK/STAT pathways by IAV infection could lead to excessive production of antiviral ISGs, resulting in decreased viral copies in vitro. However, inhibition of the JAK/STAT pathway, which regulates the expression of ISG-15, IFITM1, and TRIM22, implied that suppression of ISGs resulted in enhanced viral replication of seasonal IAVs in A549 cells. This result, however, is subject to several limitations. Only a limited number of circulating IAV strains were evaluated here; therefore, the need for continued surveillance to increase the number of samples could strengthen the concept that viral growth capability is attributable to IFN-β and antiviral ISGs response. Furthermore, the use of other strains not limited to H1N1 and H3N2, and checking different pathways leading to type I IFN production, such as TLR3 and TLR7 [36] could be explored.

The viral mechanisms involved in the suppression of IFN-β responses have been intensively investigated [37]. Most studies have associated IAV-NS1 with blocking IFN-β expression. It has been reported that NS1 point mutations at positions 45 and 66 can enhance NS1 function to counteract the interferon response [22,38]. In the present study, all strains with a high growth capability had the G45R and K66E mutations, which likely increased the viral load. Remarkably, all low-growth-capability strains have the I64T and E55X mutations that diminish the function of NS1 to counteract the effect of interferons. This can provide an unfavorable environment for replication, leading to a decreased viral load [15,21,23,24,25]. As the study is limited only to the possible contribution of NS1 mutations to the viral growth capabilities, other IAV viral components, such as RNA polymerase [39], can be elucidated for further studies.

## 5. Conclusions

In this study, we demonstrated the possible influence of ISGs on the growth capabilities of seasonal IAVs in vitro by suppressing the JAK/STAT pathway. Inhibition of ISG expression increased the viral copies of seasonal IAVs; thus, the existence of high and low growth capabilities may be caused by differences in IFN-β and ISG responses. The NS1 mutations of circulating seasonal IAVs with a low growth capability could impair IFN-β’s antagonistic ability, which can provide an unfavorable environment for replication. This evidence may contribute to the development of new and effective therapeutic strategies to reduce the risk of severe influenza infections.

## Figures and Tables

**Figure 1 vaccines-10-01507-f001:**
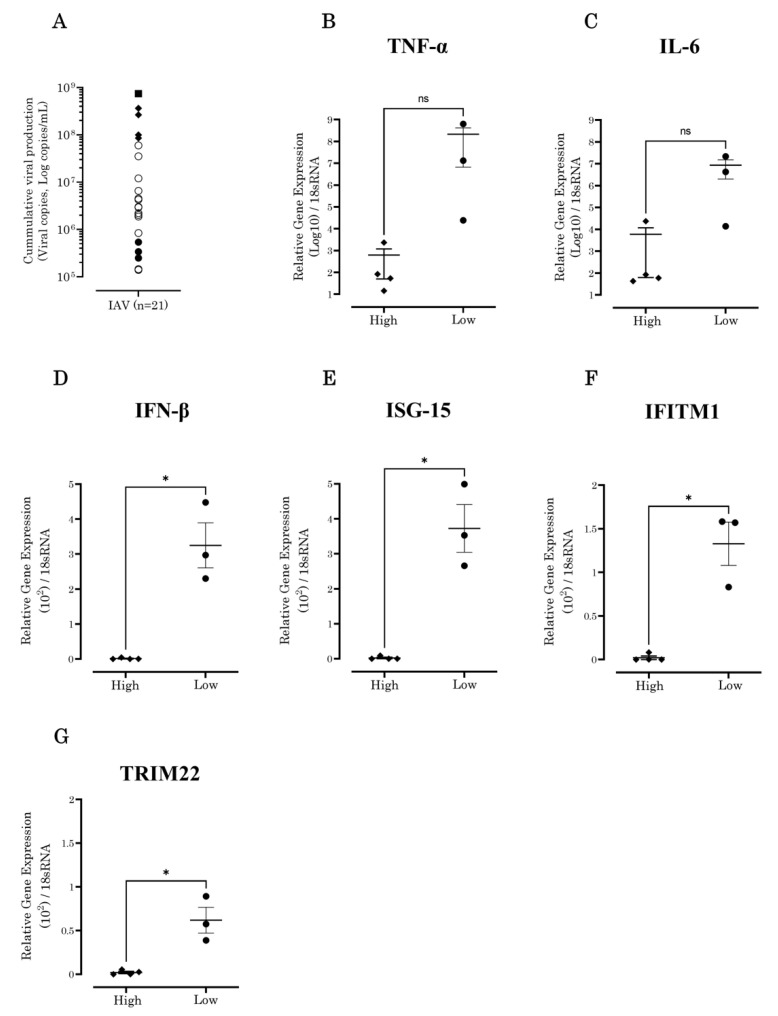
IFN-β and ISG expression levels were different in IAVs with high and low growth capabilities. (**A**) The distribution of growth capabilities is shown for 21 clinical samples in vitro using A549 cells at 7 days post-infection. The viral growth capabilities of the highest (*n* = 4) and lowest (*n* = 3) viral copy seasonal IAV strains were selected for gene expression evaluation at 6 h. Laboratory strain IAV PR8 (■) was used as positive control. (**B**) TNF-α and (**C**) IL-6 showed no significant differences when compared between high and low growth capabilities. However, (**D**) IFN-β (*p* = 0.0.0170), (**E**) ISG-15 (*p* = 0.0.0378), (**F**) IFITM1 (*p* = 0.0.0004), and (**G**) TRIM22 (*p* = 0.0.0020) were significantly different.

**Figure 2 vaccines-10-01507-f002:**
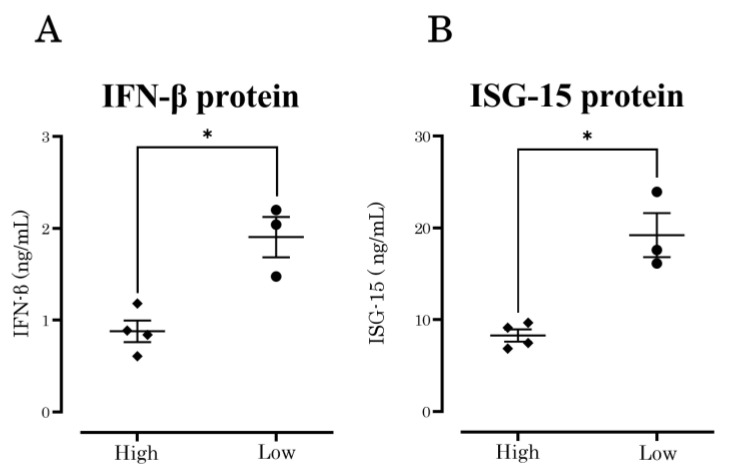
IFN-β and ISG-15 protein expression differed in IAVs with high and low growth capabilities. (**A**) IFN-β protein expression at 24 h post-infection was significantly increased (*p* = 0.0067) in IAV strains with a low growth capability compared with those with a high growth capability. (**B**) ISG-15 protein expression 24 h post-infection also showed the same pattern (*p* = 0.0038). *p* values of ≤ 0.05 is given one asterisk (*).

**Figure 3 vaccines-10-01507-f003:**
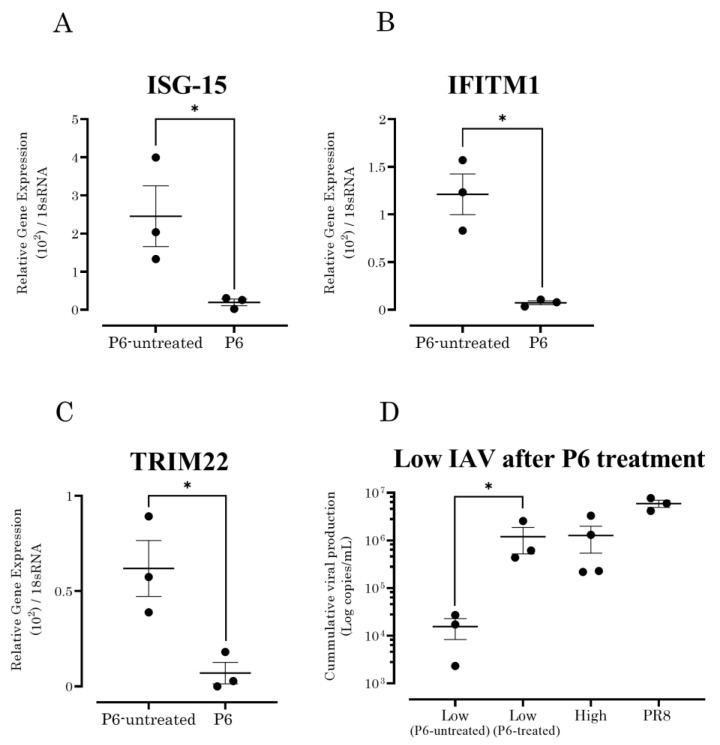
Influence of ISGs on IAVs with low growth capabilities. P6-untreated and P6-treated A549 cells were prepared to observe the JAK/STAT inhibition. Pretreatment of A549 cells with 5 μM P6, a JAK/STAT inhibitor, showed a significant suppression of (**A**) ISG-15 (*p* = 0.0479), (**B**) IFITM1 (*p* = 0.0061), and (**C**) TRIM22 (*p* = 0.0252) after 6 h post-infection of IAV strains with low growth capabilities. (**D**) P6 treatment of A549 cells inoculated with IAV strains with low growth capabilities significantly increased (*p* = 0.0483) the viral copies compared to P6-untreated cells. *p* values of ≤ 0.05 is given one asterisk (*).

**Table 1 vaccines-10-01507-t001:** List of primers used in this study.

Primers	Sequence (5′-3′)
**Virus Quantification**	
FLUA-MAT-F	CTTCTAACCGAGGTCGAAACGTA
FLUA-MAT-R	GGTGACAGGATTGGTCTTGTCTTTA
**Gene Expression**	
18S rRNA-F	GGAGCCTGCGGCTTAATTTG
18S rRNA-R	CCACCCACGGAATCGAGAAA
TNF-α-F	GCGACGTGGAACTGGCAGAAG
TNF-α-R	GGTACAACCCATCGGCTGGCA
IL-6-F	AGGATACCACTCCCAACAGACCT
IL-6-R	CAAGTGCATCATCGTTGTTCATAC
IFN-β-F	AACTGCAACCTTTCGAAGCC
IFN-β-R	TGTCGCCTACTACCTGTTGTGC
ISG-15-F	GAGAGGCAGCGAACTCATCT
ISG-15-R	CTTCAGCTCTGACACCGACA
IFITM1-F	ACTCCGTGAAGTCTAGGGACA
IFITM1-R	TGTCACAGAGCCGAATACCAG
TRIM22-F	GGGTGGACGTGATGCTGA
TRIM22-R	TCACTTGTCTCTGATCCACAGAAATA
**NS1 Sequencing**	
MBTuni-12	ACGCGTGATCAGCAAAAGCAGG
MBTuni-13	ACGCGTGATCAGTAGAAACAAGG
A-NS-M13-F	TGTAAAACGACGGCCAGTAGCAAAAGCAGGGTGACAAAGACA
A-NS-M13-R	CAGGAAACAGCTATGACCAGTAGAAACAAGGGTGTTTTTTAT

**Table 2 vaccines-10-01507-t002:** NS1 mutations present in seasonal influenza A virus that may influence the viral replication.

Influenza A Virus Subtypes	NS1 Mutations That May Influence the Viral Replication of IAV			
	Increase		Decrease	
**High Growth Capability**	**G45R**	**K66E**	**I64T**	**E55X ^a^**
A/Tottori/ST215/2009 (H1N1)	R	E	-	-
A/Tottori/ST1349/2014 (H3N2)	R	E	-	K
A/Tottori/ST488/2014 (H3N2)	R	E	-	K
A/Tottori/ST1350/2013 (H1N1)	R	E	-	K
**Low** **Growth Capability**				
A/Tottori/TT039/2011 (H1N1)	R	-	T	N
A/Tottori/ST1705/2014 (H3N2)	-	-	T	H
A/Tottori/ST777/2011 (H3N2)	-	-	T	H

^a^ Amino acid X denotes any amino acid mutation.

## Data Availability

Not applicable.

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
