# Peer review of "Pro-Inflammatory Cytokines and Interferon-Stimulated Gene Responses Induced by Seasonal Influenza A Virus with Varying Growth Capabilities in Human Lung Epithelial Cell Lines"

_vaccines, 2022, doi:10.3390/vaccines10091507_

Round 1

Reviewer 2 Report

In this manuscript, Hinay et al analyze the growth properties of seasonal influenza A virus strains that were selected from a panel of 21 nasal swabs. In particular, the authors picked viruses that grew to either high or low titers on A549 cells. All isolates induced TNF-alpha and IL-6, but only the viruses showing limited growth induced high levels of IFN-beta and ISGs. The authors show that suppression of ISG expression improves the growth of the low titer viruses, and they suggest that their sensitivity to ISG expression may be linked to specific NS1 mutations. While the authors provide no direct evidence that NS1 mutants explain the differences that they observe among the viruses, the data is suggestive and supported by previous studies. Overall, the manuscript is well-written, and the conclusions follow logically from the data. I only have two minor comments that the authors could consider to clarify their manuscript.

1.     The authors talk about “different strains” (e.g. line 161 “we obtained 4 and 3 seasonal IAV strains”), but the viruses used are really 3 isolates from the H1N1 strain and 4 isolates of the H3N2 strain. This should be made clearer in the main text.

2.     The authors may want to add to their discussion that they cannot rule out the role of other viral factors based on the current evidence. Different IAV isolates have a different innate immune activation potential, with some producing more aberrant RNA molecules than others. In addition, other viral proteins interact with components of the innate immune system as well, and there may be other mutations in addition to the NS1 mutations. There are several recent reviews on this, which the authors could cite: https://pubmed.ncbi.nlm.nih.gov/?term=influenza+innate+immune+antagonist&filter=pubt.review&sort=pubdate.

Author Response

Dear Mr or Ms. Reviewer 2

On behalf of my coauthors, I would like to thank you for the opportunity to revise and resubmit our manuscript vaccines-1851427, entitled “Suppression of Interferon Stimulated Genes Enhance Viral Replication of Seasonal Influenza A Virus through the JAK/STAT pathway”. We found the referees’ comments to be helpful in revising the manuscript and have carefully considered and responded to each suggestion. In the majority of cases, we were successful in incorporating the referees’ feedback into our revised manuscript. There were some suggestions that we were unable to implement primarily because of extension/addition of further experiments.

We have included a response to reviewers in which we address each comment the reviewers made. In our response to reviewers, the reviewers’ comments are listed in the table below together with the responses in the right column. Corresponding changes in the manuscript are marked up using the “Track Changes” function in the revised file.

Sincerely,

Kyosuke Kanai, PhD.

[email protected]

TEL: +81 589 38-6083 FAX: +81 859 38-6080

Division of Virology, Department of Microbiology and Immunology, Faculty of Medicine, Tottori University, 86 Nishi-cho, Yonago, 683-8503, Japan

List of Correction

Comments

Revisions

Reviewer 2

The authors talk about “different strains” (e.g., line 161 “we obtained 4 and 3 seasonal IAV strains”), but the viruses used are really 3 isolates from the H1N1 strain and 4 isolates of the H3N2 strain. This should be made clearer in the main text.

Added in line 168 the specific IAV subtypes A(H1N1) and A(H3N2).

The authors may want to add to their discussion that they cannot rule out the role of other viral factors based on the current evidence. Different IAV isolates have a different innate immune activation potential, with some producing more aberrant RNA molecules than others. In addition, other viral proteins interact with components of the innate immune system as well, and there may be other mutations in addition to the NS1 mutations. There are several recent reviews on this, which the authors could cite: https://pubmed.ncbi.nlm.nih.gov/?term=influenza+innate+immune+antagonist&filter=pubt.review&sort=pubdate

Added delimitations of the study on the discussion and cited references based on the referee`s suggestion.

33. Park E-S, Dezhbord M, Kim ARL and K-H. The Roles of Ubiquitination in Pathogenesis of Influenza Virus Infection. Int J Mol Sci 2022;23:4593. https://doi.org/. https://doi.org/ 10.3390/ijms23094593.

39. Elizaveta Elshina, Velthuis AJW te. The infuenza virus RNA polymerase as an innate immune agonist and antagonist. Cell Mol Life Sci 2021;78:7237–7256. https://doi.org/10.1007/s00018-021-03957-w.

Reviewer 3 Report

Hinay et. al. report circulating IAV have low and high replication potential due to the state of interferon response. Authors perform qRT-PCR assay to determine copy number of circulating IAV strains and pick 7 for further analysis. They show subsequently that high copy IAV strains have lower Type I interferon induction compared to low copy IAV. By treating A549 cells with only one pan-JAK inhibitor, authors conclude that suppression of ISG enhance viral replication of seasonal IAV through JAK/STAT pathway. I have following concerns.

MAJOR CONCERN:

Authors make a bold conclusion without substantial/robust evidence. I would like the authors to perform robust experiments before making such claim and highly recommend them to change the title of the manuscript. 

Without further experiment to show that high copy IAV strains have lower Type I interferon induction compared to low copy IAV and performing NS1mutation and rescue experiments, I won't be agreeing to publishing this manuscript. 

MINOR CONCERNS:

1. Why didn't authors use the lowest copy number IAV strain? Please clarify.
2. Please perform statistical analysis on Fig 1B. I think authors should show this data some other way as color chart doesn't do justice for such analysis.
3. Fig 2 A-C lacks gene name? 
4. Fig 2D should include positive control PR8 and high copy IAV strain.
5. Use other JAK-STAT inhibitors to perform experiments. 
6. NS1 mutation analysis is not enough to show it is the cause, you have to perform experiments to connect these dots/link.  

Author Response

Dear Mr. or Ms. Reviewer 3,

On behalf of my coauthors, I would like to thank you for the opportunity to revise and resubmit our manuscript vaccines-1851427, entitled “Suppression of Interferon Stimulated Genes Enhance Viral Replication of Seasonal Influenza A Virus through the JAK/STAT pathway”. We found the referees’ comments to be helpful in revising the manuscript and have carefully considered and responded to each suggestion. In the majority of cases, we were successful in incorporating the referees’ feedback into our revised manuscript. There were some suggestions that we were unable to implement primarily because of extension/addition of further experiments.

We have included a response to reviewers in which we address each comment the reviewers made. In our response to reviewers, the reviewers’ comments are listed in the table below together with the responses in the right column. Corresponding changes in the manuscript are marked up using the “Track Changes” function in the revised file.

Sincerely,

Kyosuke Kanai, PhD.

[email protected]

TEL: +81 589 38-6083 FAX: +81 859 38-6080

Division of Virology, Department of Microbiology and Immunology, Faculty of Medicine, Tottori University, 86 Nishi-cho, Yonago, 683-8503, Japan

List of Correction

Comments

Revisions

Reviewer 3

Authors make a bold conclusion without substantial/robust evidence. I would like the authors to perform robust experiments before making such claim and highly recommend them to change the title of the manuscript.

We proposed to change the title to:

Pro-inflammatory Cytokines and Interferon Stimulated Genes Responses Induced by Seasonal Influenza A Virus with Varying Growth Capabilities in Human Lung Epithelial Cells

Without further experiment to show that high copy IAV strains have lower Type I interferon induction compared to low copy IAV and performing NS1mutation and rescue experiments, I won't be agreeing to publishing this manuscript.

The primary goal of our study is to provide possible causes of differences in the growth capability of seasonal IAV strains. Our study intends to provide new information by profiling proinflammatory cytokines and ISGs of both high and low growth capability IAV strains. The use of JAK/STAT inhibitor is one of our supporting tools that ISGs could probably influence the differences of growth capability.

1. Why didn't authors use the lowest copy number IAV strain? Please clarify.

The strain with the lowest copy number did not produce a detectable number in RT-PCR after 6 hours.

2. Please perform statistical analysis on Fig 1B. I think authors should show this data some other way as color chart doesn't do justice for such analysis.

Change the figure from heatmap to dot graph. Statistical analysis (One-way ANOVA) comparing PR8, high replication capability strains and low replication capability strains were also performed.

3. Fig 2 A-C lacks gene name? 

Gene names were in the figure description (Line 204-205).

4. Fig 2D should include positive control PR8 and high copy IAV strain.

Added both PR8 and high copy IAV strains on Figure 2D.

5. Use other JAK-STAT inhibitors to perform experiments.

One of our references (Seng LG, Daly J, Chang KC, Kuchipudi SV. High basal expression of interferon-stimulated genes in human bronchial epithelial (BEAS-2B) cells contributes to influenza a virus resistance. PLoS One 2014;9. https://doi.org/10.1371/journal.pone.0109023) used only the same JAK/STAT inhibitor to assess the influence of ISGs to viral copies. Using similar approach, we evaluated the causes of differences in the growth capability of seasonal IAV strains in our study.

6. NS1 mutation analysis is not enough to show it is the cause, you have to perform experiments to connect these dots/link. 

We included a delimitation of the study in discussion part and mentioned that we cannot rule out the role of other viral factors and future studies can be made to further elucidation.

Round 2

Reviewer 3 Report

Dr. Kanai and authors,

Thank you for revising the manuscript. Since you retracted from making some bold conclusion as your data supports the conclusion you have made and discussed the limitations/nuances of the study, I would agree for this manuscript to be published.

Thank you!

Author Response

Thank you for your valuable commetns.